# An artificial spiking afferent nerve based on Mott memristors for neurorobotics

Xumeng Zhang [1,2,3], Ye Zhuo [1], Qing Luo[2,3], Zuheng Wu[2,3], Rivu Midya[1], Zhongrui Wang[1], Wenhao Song[1], Rui Wang[1,2,3], Navnidhi K. Upadhyay [1], Yilin Fang[2], Fatemeh Kiani[1], Mingyi Rao[1], Yang Yang[2], Qiangfei Xia [1], Qi Liu [2,3]*, Ming Liu[2,3]* & J. Joshua Yang [1]*

Neuromorphic computing based on spikes offers great potential in highly efficient computing paradigms. Recently, several hardware implementations of spiking neural networks based on traditional complementary metal-oxide semiconductor technology or memristors have been developed. However, an interface (called an afferent nerve in biology) with the environment, which converts the analog signal from sensors into spikes in spiking neural networks, is yet to be demonstrated. Here we propose and experimentally demonstrate an artificial spiking afferent nerve based on highly reliable $NbO_x$ Mott memristors for the first time. The spiking frequency of the afferent nerve is proportional to the stimuli intensity before encountering noxiously high stimuli, and then starts to reduce the spiking frequency at an inflection point. Using this afferent nerve, we further build a power-free spiking mechanoreceptor system with a passive piezoelectric device as the tactile sensor. The experimental results indicate that our afferent nerve is promising for constructing self-aware neurorobotics in the future.

---

[1] Department of Electrical and Computer Engineering, University of Massachusetts, 100 Natural Resources Road, Amherst, Massachusetts 01003, USA. [2] Key Laboratory of Microelectronic Devices and Integrated Technology, Institute of Microelectronics of the Chinese Academy of Sciences, 3 Beitucheng West Road, Beijing 100029, China. [3] University of Chinese Academy of Sciences, Beijing 100049, China. *email: liuqi@ime.ac.cn; liuming@ime.ac.cn; jjyang@umass.edu

n the era of big data and IoT, a vast amount of sensing data, such as pictures, speeches, and videos, need to be processed in real time with a high energy efficiency[1,2]. This poses a significant challenge to the traditional computing architecture due to the von-Neumann bottleneck[3–5]. Neuromorphic computing architecture based on spiking neural networks (SNNs) has been recognized as an attractive candidate for its promising energy efficiency and powerful computing capacity[6–8]. Recently, various technologies have been explored to build hardware SNNs, such as digital logic circuits[7,8], complementary metal-oxide semiconductor (CMOS) analog circuits[9,10], and emerging memristors[11–13]. Given the physical limit of transistors and their lack of desirable dynamics, memristors have attracted special attention owing to their high integration intensity[14], low power consumption[15], analog behavior[16–18], and diffusive dynamics[13,19–21], etc. Accordingly, memristor-based artificial synapses[22–24], spiking neurons[25–30], have been actively studied to construct hardware implementations of SNNs lately. However, the signals collected from the environment are usually in the continuous and analog domain, and needs to be transformed into spikes first to serve as the inputs to SNNs. Therefore, a special cell analogous to afferent nerves in biology is required to receive signals from receptors and transmit spikes to the central nervous system and brain[31,32]. Fortunately, a bio-inspired afferent nerve based on an organic ring oscillator (ORO), whose output frequency matches the action potential of the biological sensory neuron, has been reported to control the biological motor nerves by connecting to a synapse transistor[33]. The spiking frequency of the ORO could be modulated by the input voltage controlled by the pressure sensor. Then the output of the ORO was further used to trigger a synapse transistor that connected with a biological efferent nerve, in which the different output current of the synaptic transistor is converted to voltage signals to stimulate the cockroach's leg to generate different extension force. In addition, other types of devices, including two-terminal memristors and three-terminal transistors, have also been reported to emulate nociceptors[34,35], mechanoreceptors[36–38], and optical sensorimotor synapses[39], etc.,

to construct high-efficient artificial sensory systems. For these systems, a high-compact artificial spiking afferent nerve (ASAN) is needed to further transform the sensed signals into spikes. The $NbO_x$ memristor is a two-terminal device with a high integration intensity. It features negative differential resistance (NDR) behavior[40,41], which can serve as the basis of dynamic threshold switching with voltage sweeps[41] and has enabled emulation of biological neurons[26,27], and analog computing[42].

In this work, we report an artificial spiking afferent nerve (ASAN) based on a specifically designed $NbO_x$ memristor for the first time. This $NbO_x$ device is fabricated in a three-dimensional (3D) structure and with a low thermal conductivity polysilicon (poly-Si) bottom electrode to reduce the threshold current. To construct the ASAN, a compact $NbO_x$ oscillator with a $NbO_x$ device and a resistor is built first, which can transform analog input signals into correlated spiking frequencies. For such an ASAN, the input stimuli are correlated with the voltage generated by receptor devices, and the oscillation frequency is related to the spiking frequency of the neuron, which in turn depends on the intensity of the stimuli[43,44]. This ASAN shows a quasi-linear relationship between input intensity and spiking frequencies under proper stimuli, and tends to reduce firing frequency when noxious stimuli are provided, which faithfully emulates the function of biological neurons[43–45]. To further demonstrate the spiking properties of the ASAN, three types of input pulses (square, triangular, and sinusoidal) are applied, respectively. Using this ASAN, a power-free spiking mechanoreceptor system with piezoelectric device is further proposed and demonstrated experimentally. Experimental results demonstrate that our ASAN has a great potential for using in neurorobotics and can be explored to build a general afferent nerve to communicate with higher-order SNNs.

## Results

**Schematic of a spiking somatosensory system.** Figure 1 demonstrates the working principle of our afferent nerve and the

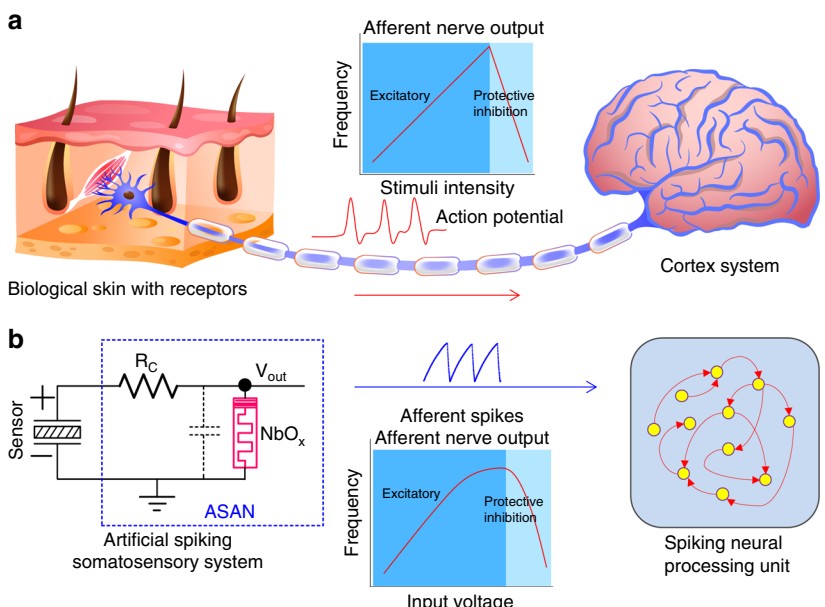

**Fig. 1 Biological afferent nerve vs. artificial afferent nerve. a** Schematic of the afferent nerve of a biological somatosensory system. Action potentials are generated in the skin and transported to the brain for processing. Spiking frequency increases with increasing stimuli intensity and decreases under strong stimuli due to the protective inhibition. **b** The artificial spiking somatosensory system, consisting of a mechanical sensor and an artificial spiking afferent nerve (ASAN) made of a resistor and a $NbO_x$ memristor. The spiking frequency shows a similar trend to that seen in its biological counterpart, which is then transmitted to the spiking neural processing unit for further processing to complete a complex task.

analogy with its biological counterpart. The bio-inspired ASAN emulates the functions of biological afferent nerves (Fig. 1a) by collecting data from somatosensory receptors and conveying this information to the cortex system. Then the cortex system will process the afferent information and transfer it to effector by the efferent nerves to respond to the external or internal environment[32]. Our artificial spiking somatosensory system (Fig. 1b) is made of a two-terminal sensor device and a compact oscillator, in which the special oscillator serves as the ASAN and contains two passive components: a resistor and a $NbO_x$ memristor. In biological systems, the firing rates of the afferent nerve increase with increasing input intensity, provided the intensity of input stimuli exceeds the threshold of the afferent nerve[43,44]. However, when the stimuli intensity is excessively high, the firing rates would decrease due to the protective inhibition of the neuron cell, to prevent the neuron from dying[45]. In our artificial somatosensory system, an analog input voltage signal is generated by the sensor device and the $NbO_x$ ASAN can transform the voltage intensity into corresponding spiking frequencies. Then the generated spikes will be transmitted to a higher-order artificial SNN for further processing. It is worth noting that the spiking frequency of the ASAN is proportional to the stimuli intensity under ordinary stimuli. Once the generated voltage intensity is overly high, the spiking frequency actually starts to decrease, just like what the biological neurons do. The quasi-linear frequency response under ordinary input intensity can be explained as the dominant and decreasing integration time of the ASAN with

increasing input intensity. The frequency decrease can be explained by considering the fact that the relaxation time of the $NbO_x$ memristor after firing becomes longer under a higher input intensity and eventually the oscillation period is mainly dominated by the relaxation time. Here we discuss two reasons that lead to the increase of the relaxation time: first, the increased input intensity leads to a higher total current flowing through the memristor after firing, which requires a longer time to discharge. Second, the increased total current induces a larger Joule heat generated in the memristor during the relaxation process, which makes the device dwell on its on-state for a longer time[46]. Once the relaxation time becomes longer than the integration time, the firing rates begin to decrease and eventually the oscillator stops firing when the device holds its on-state (see more details in Supplementary Fig. 6).

**Device characteristics and modeling.** The memristor devices used in this study are based on a 3D vertical metal–insulator–metal structure similar to those reported by us earlier[47]. The device has a titanium nitride (TiN) top electrode, a niobium oxide ($NbO_x$) switching layer, and a poly-Si bottom electrode (see Methods for the details of fabrication processes). Here, the poly-Si bottom electrode with a low thermal conductivity is specifically designed to reduce the threshold current. The cross-section transmission electron micrograph (TEM) of the device structure is shown in Fig. 2a, with the elemental mapping of the materials in the system

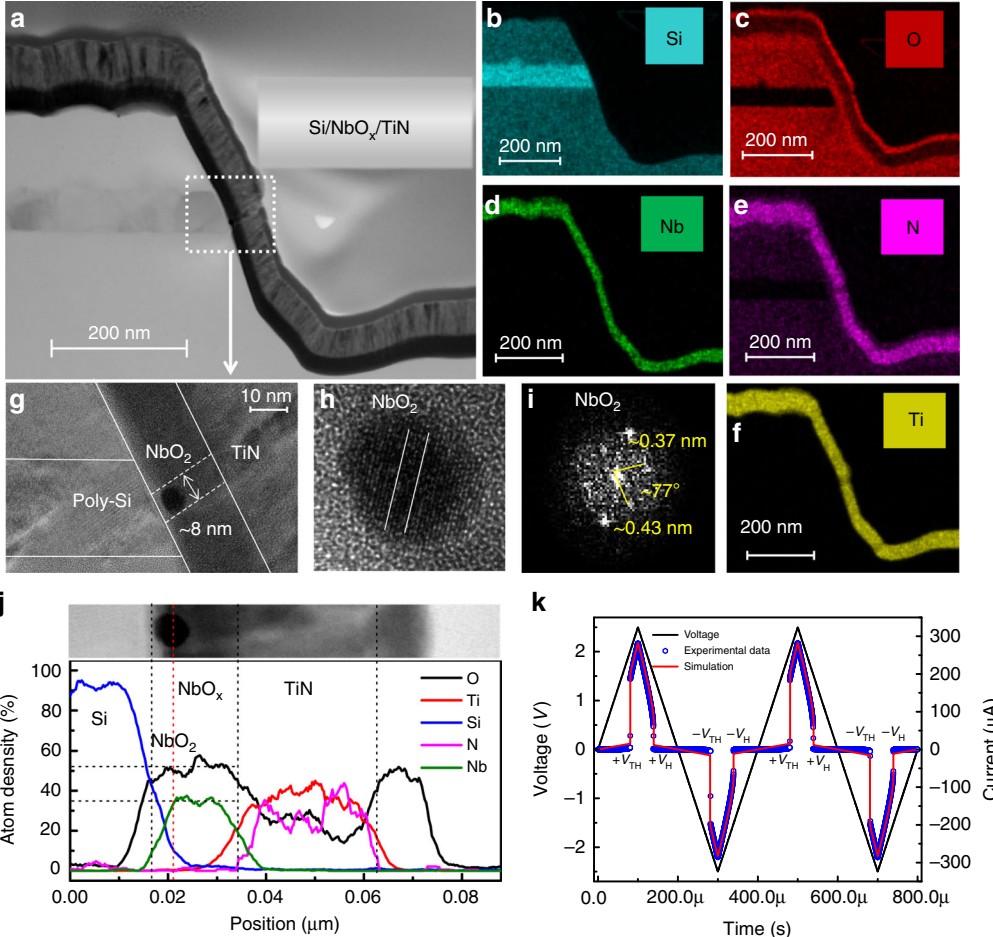

**Fig. 2 $NbO_x$ device analysis. a** Scanning electron micrograph cross-sectional image of the $NbO_x$ device. **b**–**f** The elemental mapping of the materials in the system for Si, O, Nb, N, and Ti, respectively. **g**, **h** Zoom-in views of the channel locations. **i** The diffraction pattern extracted by Fourier transform of **h**. **j** Energy dispersive spectra (EDS) of line scans of the channel. **k** Two switching cycles under triangular waves with 2.5 V/100 μs ramp rate.

for Ti, Nb, N, O, and Si (from Fig. 2b to Fig. 2f). Before operation, a DC sweep from 0 V to 5 V was required to irreversibly precondition the low-bias resistance of the devices from a 56 GΩ (@1 V) virgin state to a 35 MΩ (@1 V) operational regime (Supplementary Fig. 1a). The forming step has previously been attributed to a soft breakdown process that generates a channel of crystalline $NbO_2$ within the oxide film[41,46,48], which exhibits Joule-heating-driven NDR that underlies the threshold switching behavior[40,41,49]. During switching, ~2 μA threshold current is observed, which is much lower than devices of $NbO_2$/TiN reported previously[47,50] (see Supplementary Fig. 1). This is attributed to the low thermal conductivity of the poly-Si bottom electrode[50]. Figure 2g, h show a closer view of the channel area, a round region of $NbO_2$ crystal with a diameter of ~8 nm can be observed (Fig. 2g). We believe the round region is a cross-section of a dendrite of the $NbO_2$ channel and the complete crystalline $NbO_2$ channel might be missed during the TEM sample cutting as the TiN top electrodes were wide (~20 μm, see Methods). Figure 2h shows a zoomed-in view of the crystal $NbO_2$ region in Fig. 2g. The clear lattice fringes and the corresponding fast Fourier transform (FFT) image (Fig. 2i) indicate that the nanoparticles are highly crystalline $NbO_2$. The measured angle between relative crystal faces is ~77°, which is similar to that reported in other studies[41,48]. The elemental distributions along the channel (from Si towards TiN) are presented from the energy dispersive spectra of line scans of the cell (Fig. 2j), from which we can see that the Nb:O atoms ratio is about 2. These results suggest that a crystalline $NbO_2$ channel was formed during the forming process. After the forming operation, the device could be converted from the high resistance state (HRS) to the low resistance state (LRS) by either a positive or a negative voltage sweep. Figure 2k shows the experimental and simulation data of the device under two cycles of triangular voltage sweeps, exhibiting a bi-directional nonpolar switching behavior (details of

the model used for the simulation is provided in the Methods). In the resting state, the device is in the HRS, it switches to the LRS when the absolute value of the sweep voltage surpasses a threshold voltage ($V_{TH}$) (Fig. 2k, @~ 2.05 V) and back to the HRS when the voltage absolute value is reduced below a hold voltage value ($V_H$) (Fig. 2k, @~1.53 V). To better illustrate the switching mechanism of the device, a schematic is presented in the Supplementary Fig. 2. Initially, the switching layer is in an amorphous state (a-$NbO_x$; Supplementary Fig. 2a). After forming, a crystalline $NbO_2$ channel is generated (Supplementary Fig. 2b), which is similar to those reported in previous studies[15,41,50]. During switching operations, either a positive or a negative voltage is applied to the TiN electrode, the $NbO_2$ channel switches to the LRS when the voltage surpasses the $V_{TH}$ (Supplementary Fig. 2c); then the channel switches back to its HRS when the applied voltage falls below a hold voltage value ($V_H$) (Supplementary Fig. 2b).

**Working principle of the ASAN.** Considering the afferent nerve behavior and the characteristics of the $NbO_x$ Mott memristor, we demonstrated a compact artificial spiking afferent nerve, as shown in Fig. 3a. The ASAN is constructed with a fixed resistor ($R_c$) and a $NbO_x$ memristor, which has an intrinsic parasitic capacitance. It should be noted that the intrinsic parasitic capacitance is below one picofarad due to the nanoscale device size, rendering it negligible in comparison with the tens of picofarads of external parasitic capacitance in the test circuits. The $C_{parasitic}$ in the figure indicates the total parasitic capacitance. One node of the $R_c$ serves as the input node and another node connects with the top electrode of the $NbO_x$ memristor, with the bottom electrode grounded. The $R_c$ used here is 75 kΩ, which is much smaller than the HRS value ($R_{HRS}$) and much larger than the LRS value ($R_{LRS}$) of the $NbO_x$ memristor ($R_{HRS} \gg R_c \gg R_{LRS}$). When

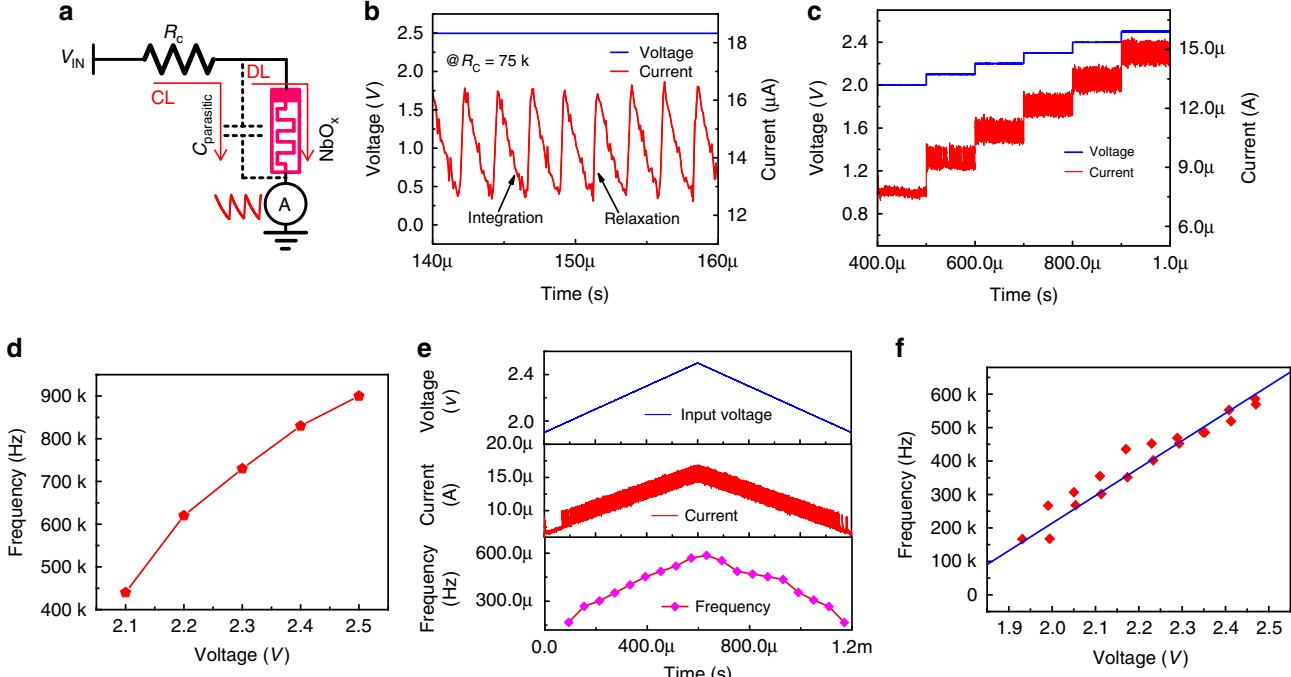

**Fig. 3 Characteristics of the artificial spiking afferent nerve (ASAN). a** Schematic of the ASAN. A resistor $R_c$ (75 kΩ) and a $NbO_x$ memristor with parallel parasitic capacitance are combined together ($R_{HRS} \gg R_c \gg R_{LRS}$). In real applications, the voltage on $NbO_x$ top electrode serves as the output spiking signal. The parasitic capacitor ($C_{parasitic}$) is tested to be about 20 pF. **b** The oscillation behavior of the ASAN. For the sake of simplicity, the current flowing through the memristor is measured as the response. The charging time from $V_H$ to $V_{TH}$ is defined as the integration time and the discharging time from $V_{TH}$ to $V_H$ as the relaxation time. **c** ASAN response under different input voltages. **d** Extracted mean value of spiking frequency vs. voltage in **c**. **e** Frequency response of the ASAN with triangular stimuli pulses. **f** The quasi-linear frequency–voltage curve extracted from **e**.

a voltage is applied on the input node, the parasitic capacitor charges through the charging loop first because $R_{HRS}C_{parasitic} \gg R_c C_{parasitic}$[29]. Once the voltage on the capacitor surpasses $V_{TH}$, the memristor device switches to its LRS due to the Joule-heating-driven NDR mechanism ($V^2/R_{HRS} \times \Delta t$) and then the capacitor discharges through the discharging loop[29]. Due to the fact that the $R_{LRS}$ is much smaller than the $R_c$ ($R_{LRS}C_{parasitic} \ll R_c C_{parasitic}$), eventually the discharging process dominates over the charging process and the net charge on the capacitor decreases. When the voltage on the capacitor falls below $V_H$, the Joule heating is not sufficient to hold the metallic state of $NbO_2$ anymore, the device returns back to its HRS, and the capacitor begins charging again. Under a continuous input, the memristor keeps switching between HRS and LRS, and an oscillation behavior can be observed, as shown in Fig. 3b. The current flowing through the memristor is measured. Here, for easy understanding, we define the charging time from $V_H$ to $V_{TH}$ as the integration time and the discharging time from $V_{TH}$ to $V_H$ as the relaxation time. It should be noted that the endurance of the threshold switch is of critical importance for practical application, so we tested the endurance by successfully running the ASAN for $\sim 10^6$ s at a period of <1 μs. After that, the device can also work normally, yielding an endurance value $>10^{12}$. (see Supplementary Fig. 3). To estimate the relationship between input intensity and output frequency, different voltages were applied on the input node (Fig. 3c). When the input voltage surpasses 2 V, the memristor starts switching and the oscillation frequency increases with increasing input voltage. Figure 3d shows the direct relationship between oscillation frequency and input voltage, each data point is the mean value under a certain voltage calculated from Fig. 3c. We can see the clear increment of the oscillation frequency with increasing voltage. High energy efficiency is known as a critical merit of biological systems. Recently, artificial synapses have been reported with a $\sim$pJ level energy consumption per spike[51,52] and even $\sim$fJ with specifically designed device based on organic materials[53,54]. To verify the feasibility of our ASAN for constructing a high-efficient artificial SNN machine, we further calculated the energy consumption of the ASAN (see Supplementary Fig. 4). Energy consumption for each spike was determined by dividing the total energy consumption by the spike numbers within a period of time, in which the total energy consumption is power integration. The minimal energy consumption as low as $\sim$38 pJ per spike event was achieved. We believe the energy consumption could be further reduced by using a $NbO_x$ device with a lower threshold voltage, a smaller $V_H$–$V_{TH}$ window, and a testing circuit with smaller parasitic capacitance[15,55]. To further demonstrate the frequency evolution under continuously increased voltage, a triangular pulse (from 1.9 V to 2.5 V) with 1 V/ms ramp was taken as the input stimuli signal, as shown in Fig. 3e. The third panel shows the corresponding frequency evolution. The oscillation frequency increases with increasing voltages and decreases with decreasing voltages. Figure 3f is the frequency–voltage relationship curve extracted from Fig. 3e. It can be concluded that the ASAN can also properly work in a triangular pulse. With our memristor model, we have also demonstrated the frequency response in simulations, a similar relationship between input intensity and spiking frequency has been observed (see Supplementary Fig. 5).

**The ASAN with an external capacitor**. To demonstrate the oscillation behavior of the ASAN under analog input sensing signal, for convenience, an external parallel 4.7 nF capacitor was used (the parallel parasitic capacitance in Fig. 3a is ~20 pF during testing process, with which the resulting resistor–capacitor time constant is not easy to be operated to complete our following artificial

mechanoreceptor system), as shown in Fig. 4a. Here, the voltage on the capacitor was measured as the output signal. Figure 4b shows the experimental results of the ASAN with a parallel capacitor (more experimental data are presented in Supplementary Fig. 6). It should be noted that the output oscillation frequency under different input voltages was tested separately using a peripheral oscilloscope. At the beginning of each input signal, an obvious integration process of the capacitor can be observed. Then the incremental input voltages were applied successively, the output voltage oscillates between $V_{TH}$ and $V_H$. The frequency–voltage curve is shown in Fig. 4c. When the input voltage increases from 2.6 V to 4.8 V, the oscillation frequency shows a quasi-linear increment. Then the spiking frequencies decrease with further increasing input voltage and eventually stops oscillating at 6.2 V due to the fact that the memristor start to hold its on-state. The behavior of increasing frequency with increasing input intensity has been systematically demonstrated in previous literature[55,56], whereas the frequency decrease with further increasing input intensity has not been specifically reported yet. The quasi-linear increment of frequency under ordinary input intensity can be explained as the dominant and decreasing integration time of the ASAN with increasing the input intensity (spike period = integration time + relaxation time (see Supplementary Fig. 7a, b). Furthermore, owing to the increasing input intensity, the total current flowing through the memristor is increased within the relaxation process (see Fig. 3c and simulation results in Supplementary Fig. 8), which requires a longer time to discharge. In addition, the increasing current flow through the memristor leads to more Joule heat in the memristor during the relaxation process, which makes the device dwell on its on-state for a longer time. Consequently, the relaxation time of the ASAN increases and eventually dominates the oscillation period (see Supplementary Fig. 7d). Thus, the spiking period time of the ASAN decreases first and then increases, and correspondingly the frequency increases first and then decreases. It should be noted that the ASAN stops firing continuously under a higher voltage owing to the voltage divided on $NbO_x$ device is always larger than its holding voltage after the first-fire event. The voltage on the device could generate sufficient Joule heat to hold the device in its "on-state" (see Supplementary Fig. 9). This frequency response is similar to the biological afferent nerve whose spiking rates increase with an increase in the intensity of harmless stimuli, but decreases and eventually stops firing under excessively strong stimuli due to the intrinsic protective-inhibition mechanism that serves to keep the system balanced and prevent neurons from dying[45].

In nature, signals are usually transmitted in an analog form and the generated signals from sensors are continuous[57]. To mimic such signals, sinusoidal signals with or without DC bias were applied on the input node, as shown in Fig. 4d, e. The input sinusoidal signal with DC bias (makes the input signal only with positive voltage) was applied first (Fig. 4d). The output spiking frequency is presented in the third panel. We can see that the frequency–time curve also exhibited a sinusoidal form. The spiking behavior generated by a sinusoidal input signal without DC bias (with both positive and negative voltage) is given in Fig. 4e. The ASAN can be successfully operated, and the corresponding frequency displays the same evolutionary trend as the case with only positive input voltage. Figure 4f shows the output spiking behavior under a strong input voltage. The protective inhibition of neuron cells is observed and could recover to the excitable state whenever the input voltage becomes normal. We note that the stop spiking voltage in Fig. 4f is about 5.9 V, which is slightly smaller than that in Fig. 4c (6.2 V). This difference results from the $V_H$ variability of $NbO_x$ device. The $V_H$ at the moment of testing Fig. 4f (1.62 V) is lower than that in Fig. 4c (1.71 V); thus, a lower stop spiking voltage in Fig. 4f is

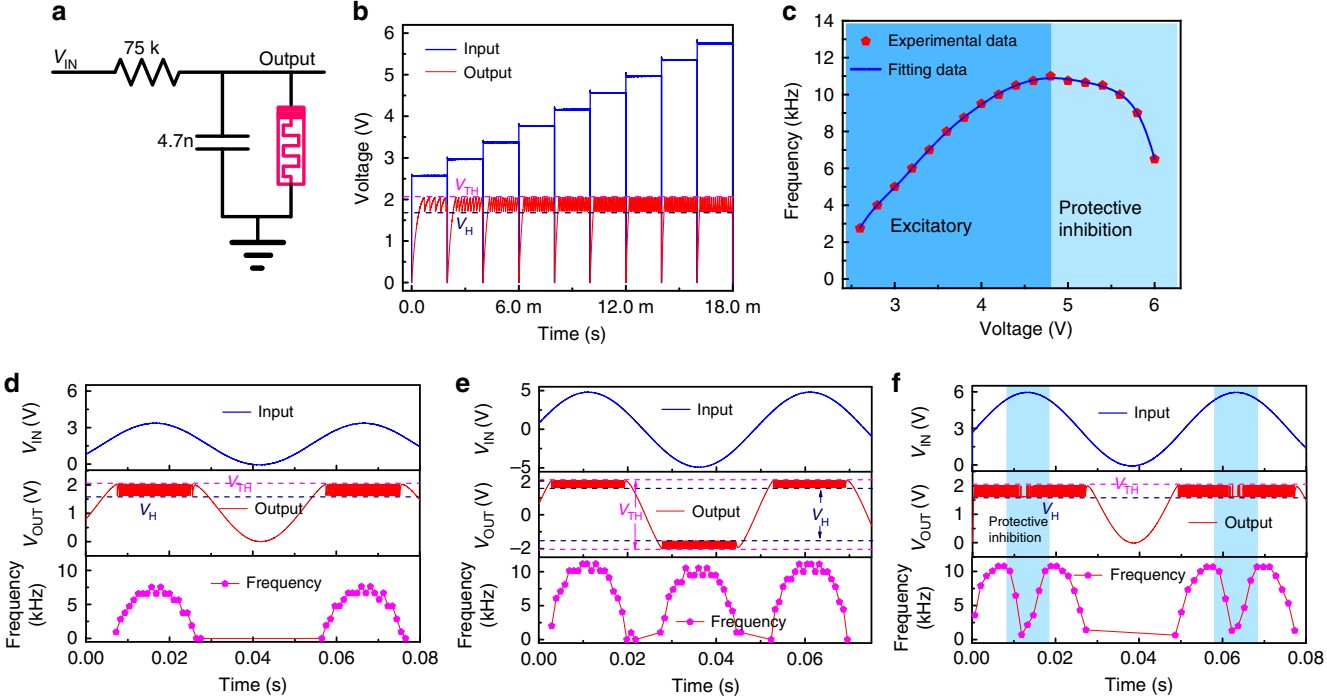

**Fig. 4 Artificial spiking afferent nerve (ASAN) with an external capacitor. a** Schematic of the ASAN with an external parallel capacitor (4.7 nF). **b** Frequency response with different input voltages. **c** The frequency–voltage curve with two stages: excitatory spiking stage under low input voltages and protective inhibition stage under high voltages. **d**–**f** The frequency response with a sinusoidal signal as input. **d**, **f** The input signals only with positive voltage, protective inhibition can be observed in **f** which has a higher amplitude. **e** An input sinusoidal signal without bias, where the oscillation behavior can be obtained upon applications of both positive and negative voltage.

observed. Besides, to extend the application of our ASAN, we employed a 47 nF capacitor to test its spiking behavior and achieved a lower frequency range from 0 Hz to 1100 Hz, which matches the human nervous system (from 1 Hz to 1000 Hz) (see Supplementary Fig. 10). These results demonstrated that our afferent nerve could work with analog signals and has a great potential to be used in various environments, even suitable for the human–machine interface.

**Power-free artificial spiking mechanoreceptor system.** Mechanoreceptors are primary sensory structures for detecting mechanical stimuli (e.g., pressure, touch, stretching, and vibration) and for sending the generated responses to the brain by afferent nerve for further processing to generate an appropriate response to the external and internal environments. Under harmless stimuli, mechanoreceptors can encode mechanical deformation into proportional spiking frequency[37] but tend to stop firing under noxious stimuli due to the protective inhibition mechanism[45]. With a piezoelectric device connecting to the ASAN, we demonstrated an artificial spiking mechanoreceptor system (ASMS) (Fig. 5a). When pressure was applied on a piezoelectric device, a positive voltage was generated on the top electrode and a negative voltage was generated when pressure was lifted[58,59]. The positive and negative voltages are a result of the deformation of atomic structures[59]. It should be noted that the input voltage signal of our artificial mechanoreceptor is generated by the piezoelectric device, so the system does not need an external power source. The piezoelectric device started to generate a voltage at the beginning of the time when the pressure was applied and then up to a peak value depending on the pressure intensity, and then the voltage decreases due to the leaky nature of the charge[59]. An opposite voltage was generated when the finger was lifted and then the charge leaked (see Supplementary Fig. 11). Figure 5b shows the experimental results of the artificial mechanoreceptor system.

When a force is applied on the piezoelectric device, a sinusoid-like voltage signal was generated and then this signal was transformed into spiking signals by the ASAN. It can be seen that the ASAN exhibits the same spiking behavior as it did when an analog input voltage from an external power source was applied to it. When the force is high, a high peak voltage is generated, which makes the ASAN stop spiking (protective inhibition), as shown in Fig. 5c (a closer view of Fig. 5b). Figure 5d–f show the zoom-in of the other time slot of Fig. 5b. The dynamic frequency response can be clearly observed, which has the same trend with the generated voltage. To illustrate the frequency response of our ASMS under different pressure, we apply different forces of varying intensities on the piezoelectric device, as shown in Fig. 5g. When the pressure force on the piezoelectric device is small, the generated voltage is insufficient to drive the memristor, then no dynamic spiking behavior can be obtained. Once the applied pressure force is sufficiently large, the voltage generated by the piezoelectric device can drive the memristor to switch. The peak frequency of the afferent nerve increases with increasing the pressure force; more experimental data under other pressure intensity are presented in Supplementary Fig. 12. These results demonstrate that a power-free artificial mechanoreceptor has been successfully implemented experimentally and the afferent nerve can be used for transforming analog sense signals into dynamic spiking frequencies. These results suggest that our afferent nerve has a great potential to be used in spiking neurorobotics.

## Discussion
Neuromorphic machines consisting of spiking neurons and synapses could provide a more efficient approach to performing complex tasks than traditional hardware. An ASAN combined with sensors is critical for interacting with the environment, which converts the analog signal in the environment into spiking signals that could be further processed by the neuromorphic

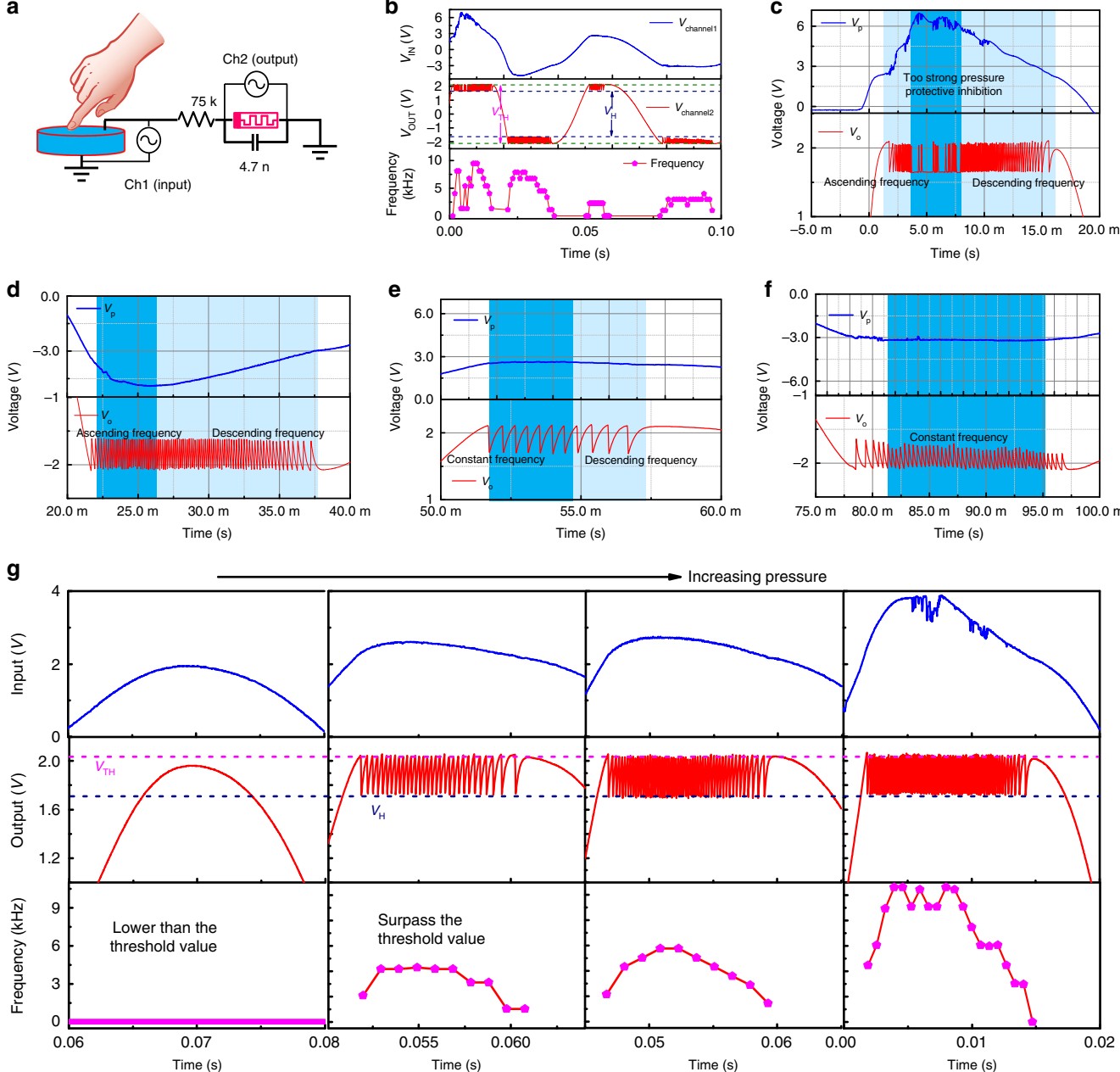

**Fig. 5 Illustration of the artificial spiking mechanoreceptor system (ASMS). a** Schematic of the ASMS. A piezoelectric device is used as the tactile sensor and connected with the artificial spiking afferent nerve. The voltage generated by the piezoelectric device serves as the input signal. **b** The experimental data of the ASMS. **c–f** A closer view of **b**, the protective inhibition behavior can be observed in **c**. **g** The frequency response of the ASMS under different pressure intensities.

machine. The ASAN using CMOS ring oscillators shows the analog information could be converted into spiking signals[33,37], in which the spiking frequency is dominated by the inverter delay. Given the physical limitations of transistors and their lack of desirable dynamics, oscillators with memristors become a more promising candidate, owing to their high integration intensity, low power consumption, and inherent dynamics, etc. However, previous works merely focus on the emulation of dynamic cortex neurons or operations for classifications[13,26,27,56]. The interface between the neuromorphic machine and the environment is also a critical component to construct a self-aware machine.

In summary, we have proposed and experimentally demonstrated an ASAN based on specially engineered NbO$_x$ Mott memristors. The ASAN can transform analog signals into

dynamic oscillation frequencies. The frequency has a quasi-linear relationship to the input voltage within a certain range of the input signal intensity and tends to stop spiking under noxious input intensity, closely resembling a biological neuron. The dynamic spiking behavior under various input signals, such as rectangular, triangular, and sinusoidal pulses, was studied systematically. We further integrated the ASAN into a piezoelectric device to construct an ASMS without any external power source. The ASMS can respond to the pressure signal and transform the pressure intensity into a corresponding spiking frequency. The ASAN can be readily extended to process sensory signals from other sensors, such as smell, taste, sight, hearing, temperature, magnetic field, and humidity. In addition to constructing the diversiform sense systems, our nerve cell is also suitable for

applications in spiking neurons owing to its leaky integration and fire characteristics, or coupled oscillator neural network owing to its input-intensity-dependent oscillation behavior[56,60]. The nerve can thus be further used to construct complex neural networks to process central information and fabricate a highly efficient spiking neurorobotics system.

## Methods

**Device fabrication.** First, one $SiO_2(150\,nm)/Si(60\,nm)/SiO_2(150\,nm)$ multi-layers were deposited by PVD and PECVD, respectively. Patterning and only one-step etching were applied to form the bottom electrode (poly-Si) with a smooth sidewall profile. Then the $NbO_x$ switching layer (~25 nm) and TiN top electrode (~40 nm) were deposited on the sidewalk sequentially by magnetron sputtering at room temperature, followed by lift-off process to form top electrodes. The area of the memristor cell is defined by the thickness of the bottom electrode Si (60 nm) and the width of the literal top electrode (TiN) width (20 μm).

**Measurement method.** In the electrical experiment, the electrical characteristics of a single $NbO_x$ device and the experimental results in Fig. 3 are performed on an Agilent B1500A. During the test with an external parallel capacitor (Fig. 4), a Keysight 81160A pulse generator is used as the power source to generate input signals and a Keysight InfiniiVision MSO-X 3104T oscilloscope is performed to measure the input signal and output spikes. The artificial mechanoreceptor system, in which the tactile sensor is implemented using off-the-shelf piezoelectric device, and a Keysight InfiniiVision MSO-X 3104T oscilloscope are used to measure the generated input and output signals (Fig. 5).

**LTspice device model.** In this study, we used a biphasic memristor model proposed in ref. [15]. In this model, there are four assumptions: (a) cylindrical symmetry, (b) constant temperature within the metallic core fixed at the transition temperature, (c) ambient temperature at the exterior of the channel, and (d) two-dimensional heat flow along the radial direction. The total device resistance is described by a function of phase fraction (Eq. 1):

$$R_{ch}(u) = \frac{\rho_{ins}L}{\pi r_{ch}^2}\left[1 + \left(\frac{\rho_{ins}}{\rho_{met}} - 1\right)u^2\right]^{-1} \quad (1)$$

where $R_{ch}$ is the channel resistance, $\rho_{ins}$ and $\rho_{met}$ are metallic and insulating phase electrical resistivity, respectively, $r_{ch}$ is the conduction channel radius, $L$ is the channel length $u = r_{met}/r_{ch}$ is the metallic phase fraction expressed in radial coordinates. The dynamical sate evolution relation with time $t$ is presented as the Eq. 2:

$$\frac{du}{dt} = \left(\frac{d\Delta H}{du}\right)^{-1}\left(R_{ch}(u)i^2 - \Gamma_{th}(u)\Delta T\right) \quad (2)$$

In Eq. 2, the $\Delta H$ and $\Gamma_{th}$ is the system enthalpy and thermal conductance of the insulating shell, respectively, and $\Delta T$ is the heating temperature; they could be presented as Eqs. 3 and 4:

$$\Delta H = \pi L r_{ch}^2\left[c_p\Delta T\frac{u^2-1}{2\ln u} + \Delta h_{tr}u^2\right] \quad (3)$$

$$\Gamma_{th}(u) = 2\pi L\gamma\left(\ln\frac{1}{u}\right)^{-1} \quad (4)$$

The variation of the enthalpy with respect to $u$ is thus as the Eq. 5

$$\frac{d\Delta H}{du} = \pi L r_{ch}^2\left[c_p\Delta T\frac{1-u^2+2u^2\ln u}{2u(\ln u)^2} + 2\Delta h_{tr}u\right] \quad (5)$$

According to the above five equations, we programmed a LTspice model and carried out the simulation.

## Data availability

All data needed to evaluate the conclusions in the paper are present in the paper and/or the Supplementary Materials. Additional data related to this paper may be requested from the authors.

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

## Acknowledgements

X.Z., Q. Luo, Z.H.W., R.W., Y.F., Y.Y., Q. Liu and M.L. were supported by the National High Technology Research Development Program under Grant Number 2017YFB0405600; the National Natural Science Foundation of China under Grant Numbers 61825404, 61732020, 61821091, 61804167, and 61851402; Major Science and Technology Special Project of China under Grant Number 2017ZX02301007–001; and the Strategic Priority Research Program of the Chinese Academy of Sciences under Grant Number XDPB12. Y.Z., R.M., Z.R.W., W.S., N.K.U., F.K., M.R., Q.X. and J.J.Y. were supported in part by the US Air Force Research Laboratory (AFRL) (Grant Number 12 FA8750–18–2–0122) and by Aire Force of Scientific Research (AFOSR) for the supported through the MURI program under contract number FA9550–19–1–0213. Any opinions, findings, and conclusions or recommendations expressed in this material are those of the author and do not necessarily reflect the views of AFRL.

## Author contributions

X.Z. and J.J.Y. designed the experiments. X.Z. carried out the electrical experiments. X.Z. and Y.Z. conducted the simulation. X.Z., Q. Luo, Q. Liu and M.L. designed and fabricated the Mott devices. X.Z., Z.H.W., Y.F., Y.Y. and Q. Liu carried out the TEM and energy dispersive spectra test. R.M., Z.R.W., W.S., R.W., N.K.U., F.K., M.R. and Q.X. helped with data analysis. X.Z., J.J.Y., Q. Liu and M.L. prepared the paper. Q. Liu, M.L. and J.J.Y. supervised the research.

## Competing interests

The authors declare no competing interests.
