## [Peer Review File · Nature Communications]

Reviewers' comments:

Reviewer #1 (Remarks to the Author):

This is a report on the fabrication of an artificial spiking afferent nerve (ASAN) using NbOx Mott memristors as a basic component. The artificial spiking mechanoreceptor system based on ASAN could be potentially used in a self-aware neuro-robotics. The results are interesting. Acceptance is suggested after the following issues can be addressed.

1) The working frequency in human nervous system is usually between 1-1000 Hz. Matching of the frequency range could extend the applications of the ANSN to human-machine interface, which could be another benefit of the system. Could the authors provide more data in this frequency range if possible?

2) Protective inhibition behavior has been discussed in Figure 5. How large voltage can the device be sustainable? Have the authors tested if the device can maintain this special protective inhibition characteristic under the higher working voltage, e.g. 8V or 10V? What is the breakdown voltage of the NbOx memristor?

3) A very important superiority of nervous system is its high energy efficiency. Could the authors provide the information on the energy consumption of ASAN? Correlated papers regarding energy consumption of synaptic devices are suggested to be cited and discussed (e.g. Science advances 2 (6), e1501326 (2016); Advanced Materials 28 (28), 5916-5922 (2016); Materials Chemistry Frontiers 3 (5), 941-947 (2019); Nanotechnology 30 (1), 012001 (2018)).

4) Artificial afferent nerve has already been demonstrated in the year 2018 using transistor-structured synaptic devices, and recently, great efforts have been made in this research field. Correlated works need to be cited in the introduction part with detailed discussions to let readers understand current research status. (e.g. Science 360 (6392), 998-1003 (2018); Small 15 (32), 1970170 (2019))

Reviewer #2 (Remarks to the Author):

In the manuscript titled "An artificial spiking afferent nerve based on Mott memristor for neuro-robotics", the authors investigated artificial spiking afferent nerve based on Mott memristors. The stimuli intensity was converted to spike frequency via artificial spiking somatosensory system. To show the feasibility of the system, the author used a piezoelectric device to generate input voltage and showed the frequency response of the system under different pressure intensities. The reviewer believes that researchers in this field will be very interested in the contents of this manuscript. Therefore, the reviewer would recommend the publication of this paper to Nature Communication after the following minor revision. These comments are not to criticize but to strengthen the manuscript.

1. Relaxation state can be thought of as a state with an open circuit that has no inputs. However, in the manuscript, current still flows in the relaxation process. The reviewer recommends to explain what each state means in the manuscript.

2. What is the device's intrinsic parasitic capacitance?

3. The reviewer recommends to add recently published "Science Advances Vol. 4, no. 11, eaat7387" as a reference which could be related to the article.

4. In reviewer's opinion, it might be better to add external parasitic capacitor's capacitance in Fig 3. or in the caption.

5. The reviewer recommends to plot current that flows at memristor and calculate joule heat value

in case of the large input voltage. Moreover, please check whether the value of joule heat is enough to make ASAN dwell on its "on-state".

6. It seems that, when the input voltage is large, the spiking frequency descends initially and stop spiking. But when dose the ASAN stops spiking? In Fig 4c. it says that when the voltage is 6 V, the frequency remains in about 6kHz. However, in Fig 4f., the ASAN stops spiking when the voltage is 6 V, which should have the frequency of 6kHz according to Fig 4c. Please describe the occasion that ASAN stops spiking in Fig 4. or in the manuscript.

7. Please check over some errors due to a typo. In page 11 line 265, the word "verve" should be changed to "nerve".

We thank all the reviewers for their precious time and constructive comments on our manuscript. We revised both the main text and the supplementary information (SI) accordingly.

The changes we made are summarized as follows:

1. We provided the data of the ASAN works on a lower frequency (0-1100 Hz) in Supplementary Fig. S10 and added one sentence to describe the ASAN could apply to the human-machine interface in the main text.
2. We provided the energy consumption of the ASAN in the case of Fig. 3c and added Supplementary fig. S4 to demonstrate it.
3. Added more introduction about the artificial afferent nerve and cited published papers on page 2 and page 3 in the main text.
4. Added the declaration of the integration time and relaxation time on page 7 in the main text.
5. We calculated the flowing current and power consumption of the NbO_x device in ASAN under a higher input voltage and explained why ASAN dwelling on its “on-state”, the results are shown in the added Supplementary fig. S9.
6. Added the explanation for the difference of stop spiking voltages between Fig. 4c and Fig. 4f on page 11.
7. We have revised the typo “verve” on page 12 to “nerve”, and the typo “however” on page 1 to “However”.

In the following point-to-point response, the original comments are in black fonts, and our responses are in blue fonts. Changes in the revised main text and SI are highlighted in yellow color.

One by one Response to Reviewers' comments:

Reviewer #1 (Remarks to the author)

This is a report on the fabrication of an artificial spiking afferent nerve (ASAN) using NbO_x Mott memristors as a basic component. The artificial spiking mechanoreceptor system based on ASAN could be potentially used in a self-aware neuro-robotics. The results are interesting. Acceptance is suggested after the following issues can be addressed.

Response: We thank the referee for the positive comments on the significance of our work. Our responses to the comments one by one are shown as follows.

Comments:

1) The working frequency in human nervous system is usually between 1-1000 Hz. Matching of the frequency range could extend the applications of the ASAN to human-machine interface, which could be another benefit of the system. Could the authors provide more data in this frequency range if possible?

Response: We thank the referee for this helpful suggestion. We agree with the referee that matching the human frequency range could extend the applications of the ASAN to the human-machine interface. As we mentioned in the main text, the spiking frequency of our ASAN depends on the sum of the integration time and the relaxation time, which is determined by the RC constant. To demonstrate this application, a bigger capacitor (47 nF) is used. The tested data is shown in Fig. R1. The results demonstrate that our ASAN could also work in a lower frequency range that matching the human nervous system (0-1100 Hz). To declare this application, we added one sentence in the main text “**Besides,**

to extend the application of our ASAN, we employed a 47 nF capacitor to test its spiking behavior and achieved a lower frequency range from 0 Hz to 1100 Hz which matches the human nervous system (from 1 Hz to 1000 Hz) (see fig. S10). These results demonstrated that our afferent nerve could work with analog signals and has a great potential to be used in various environments, even suitable for the human-machine interface.”

Fig. R1. The spiking behavior of the ASAN with a 47 nF capacitor. (a) Schematic of the test circuits. (b) The tested data of the input voltages, output spikes, and corresponding current signal flowing through the NbO_x device. (c) Zoom-in views of (b). (d) The spiking frequency as a function of input voltages. A spiking range from 0 to 1100 Hz matching the human nervous system (1~1000 Hz) is obtained.

2) Protective inhibition behavior has been discussed in Figure 5. How large voltage can the device be sustainable? Have the authors tested if the device can maintain this special protective inhibition characteristic under the higher working voltage, e.g. 8V or 10V? What is the breakdown voltage of the NbO_x memristor?

Response: We thank the referee for this comment. As we demonstrated in the main text, during working, the voltage on the NbO_x device is between V_H and V_{TH} . To test how large voltage the device can sustain, we performed the I-V test without compliance current, as shown in Fig. R2a. Compared to the I-V curve with compliance current, the device can normally work up to a 2.7 V voltage. This is because of the self-compliance current characteristics of the NbO_x device. When we increased the voltage to 3.7 V, the device still performed a threshold switching behavior, but the V_{TH} and V_H show an obvious shift. This is because a larger voltage generated more Joule heat on the device and affected the state of the initial NbO₂ channel. The threshold switching behavior disappeared when a 3.8 V voltage applied to the device, as shown in Fig. R1b. In this condition, the device cannot support the normal spiking behavior of the ASAN.

We have also tested the ASMS under a higher stress which generates an ultimate voltage about 8V, the device still maintains the protective inhibition characteristic, as shown in Fig. R2c. This is because the actual voltage applied to the device is still between V_H and V_{TH} .

We think the breakdown voltage could be thought of a voltage that destroys the normal threshold switching behavior of the device. For this measurement, the breakdown voltage is 3.8 V (Fig. R2b). Besides, we also performed the voltage sweep under a higher voltage, as shown in Fig. R2d. When the applied voltage reaches 6.3 V, the device is reset to a

high resistance state due to the breakdown of NbO₂ channel under the Joule-heating effect. Followed by a forming process and presents the memory behavior due to the lack of a current compliance.

Fig. R2. (a) The I-V curves of the device under small voltages with no compliance current. (b) Increasing the sweep voltage, the V_H and V_{TH} show an obvious shift. And the threshold switching behavior disappears when a 3.8 V voltage applied to the device. (c) The spiking behavior under a higher input voltage of the ASMS. The ultimate voltage generated by the piezoelectric device is about 8 V, in which the device can still work well. In this condition, the actual voltage applied to the device is still between V_H and V_{TH} . (d) When the applied voltage reaches 6.3 V, the device is reset to a high resistance state due

to the breakdown of NbO₂ channel under the Joule-heating effect. Followed by a forming process and presents the behavior of memory due to the lack of a current compliance.

3) A very important superiority of nervous system is its high energy efficiency. Could the authors provide the information on the energy consumption of ASAN? Correlated papers regarding energy consumption of synaptic devices are suggested to be cited and discussed (e.g. Science advances 2 (6), e1501326 (2016); Advanced Materials 28 (28), 5916-5922 (2016); Materials Chemistry Frontiers 3 (5), 941-947 (2019); Nanotechnology 30 (1), 012001 (2018)).

Response: We thank the referee for this comment. We agree with the referee that high energy efficiency is a very important superiority of the nervous system. To present this, we calculated the energy consumption for each spike signal in Fig. 3c, as shown in Fig. R3. The results demonstrated that the minimal energy consumption could be as low as ~ 38 pJ per spike event. In addition, we believe the energy consumption could be further reduced by using a NbO_x device with a lower threshold voltage, a smaller V_H-V_{TH} window, and a testing circuit with smaller parasitic capacitance^{R1,R2}.

Correspondingly, we add descriptions in the main text. “High energy efficiency is known as a critical merit of biological systems. Recently, artificial synapses have been reported with a ~pJ level energy consumption per spike^{51,52} and even ~ fJ with specifically designed device based on organic materials^{53,54}. To verify the feasibility of our ASAN for constructing a high-efficient artificial SNN machine, we further calculated the energy consumption of the ASAN (see fig. S4). Energy consumption for each spike was

determined by dividing the total energy consumption by the spike numbers within a period of time, in which the total energy consumption is power integration. The minimal energy consumption as low as ~ 38 pJ per spike event was achieved. We believe the energy consumption could be further reduced by using a NbO_x device with a lower threshold voltage, a smaller V_H - V_{TH} window, and a testing circuit with smaller parasitic capacitance^{15,41}.

The suggested papers are added in the corresponding locations, and the number of the references is updated accordingly in the revised manuscript.

51. H. Yu, J. Gong, H. Wei, W. Huang W, and W. Xu. Mixed-halide perovskite for ultrasensitive two-terminal artificial synaptic devices. *Mater.Chem. Front.* **3**, 941-947 (2019).
52. Y. Chen, H. Yu, J. Gong, M. Ma, H. Han, H. Wei, and W. Xu, Artificial synapses based on nanomaterials, *Nanotechnology* 30, 012001 (2019).
53. W. Xu, S. Y. Min, H. Hwang, and T. W. Lee, Organic core-sheath nanowire artificial synapses with femtojoule energy consumption. *Sci. Adv.* **2**, 7 (2016).
54. W. Xu, H. Cho, Y.-H. Kim, Y.-T. Kim, C. Wolf, C.-G. Park, and T.-W. Lee, Organometal Halide Perovskite Artificial Synapses, *Adv. Mater.* **28**, 5916-5922 (2016).

Fig. R3. The energy consumption of the ASAN for each spike under different frequency. (a)-(e) The energy consumption is extracted from Fig. 3c. The transient power is calculated by the multiply of input voltage and output current, and the energy consumption for each spike was calculated by dividing the total consumption by the spike numbers. (f) The energy consumption per spike under different spiking frequency.

4) Artificial afferent nerve has already been demonstrated in the year 2018 using transistor-structured synaptic devices, and recently, great efforts have been made in this research field. Correlated works need to be cited in the introduction part with detailed discussions to let readers understand current research status. (e.g. Science 360 (6392), 998-1003 (2018); Small 15 (32), 1970170 (2019))

Response: We thank the referee for this comment. It is a great suggestion to introduce more details in this part and let readers better understand the current research status.

We added descriptions in the introduction part. “Fortunately, a bio-inspired afferent nerve based on an organic ring oscillator (ORO), whose output frequency matches the action

potential of the biological sensory neuron, has been reported to control the biological motor nerves by connecting to a synapse transistor³³. The spiking frequency of the ORO could be modulated by the input voltage controlled by the pressure sensor. Then the output of the ORO was further used to trigger a synapse transistor that connected with a biological efferent nerve, in which the different output current of the synaptic transistor is converted to voltage signals to stimuli the cockroach's leg to generate different extension force. In addition, other types of devices, including two-terminal memristors and three-terminal transistors, have also been reported to emulate nociceptors^{34,35}, mechanoreceptors^{36,37} and optical sensors³⁸ et al., to construct high-efficient artificial sensory systems. For these systems, a high-compact artificial spiking afferent nerve is needed to further transform the sensed signals into spikes. The NbO_x memristor is a two-terminal device with a high integration intensity. It..."

The suggested papers are added in the corresponding locations and the number of the references is updated accordingly in the revised manuscript.

33. P. D. Wall and M. Gutnick, Properties of afferent nerve impulses originating from a neuroma, *Nature* **248**, 740-743 (1974). Y. Kim, A. Chortos, W. T. Xu, Y. X. Liu, J. Y. Oh, D. Son, J. Kang, A. M. Foudeh, C. X. Zhu, Y. Lee, S. M. Niu, J. Liu, R. Pfattner, Z. N. Bao, and T. W. Lee, A bioinspired flexible organic artificial afferent nerve, *Science*, **360**, 998-1003 (2018)
34. Y. Kim, Y. J. Kwon, D. E. Kwon, K. J. Yoon, J. H. Yoon, S. Yoo, H. J. Kim, T. H. Park, J. W. Han, K. M. Kim, and C. S. Hwang, Nociceptive memristor, *Adv. Mater.* **30**, 1704320 (2018).
35. J. H. Yoon, Z. Wang, K. M. Kim, H. Wu, V. Ravichandran, Q. Xia, C. S. Hwang, and J. J. Yang, An artificial nociceptor based on a diffusive memristor, *Nat. Commun.* **9**, 417 (2018).

36. H. Han, H. Yu, H. Wei, J. Gong, and W. Xu. Recent Progress in Three-Terminal Artificial Synapses: From Device to System. *Small* **15**, 1900695 (2019).
37. B. C. K. Tee, A. Chortos, A. Berndt, A. K. Nguyen, A. Tom, A. McGuire, Z. L. C. Lin, K. Tien, W. G. Bae, H. L. Wang, P. Mei, H. H. Chou, B. X. Cui, K. Deisseroth, T. N. Ng, and Z. N. Bao, A skin-inspired organic digital mechanoreceptor, *Science* **350**, 313-316 (2015).
38. Y. Lee, J. Y. Oh, W. Xu, O. Kim, T. R. Kim, J. Kang, Y. Kim, D. Son, J. B.-H. Tok, M. J. Park, Z. Bao, and T.-W. Lee, Stretchable organic optoelectronic sensorimotor synapse, *Sci. Adv.* **4**, 7387 (2018).

Reviewer #2 (Remarks to the author)

In the manuscript titled "An artificial spiking afferent nerve based on Mott memristor for neuro-robotics", the authors investigated artificial spiking afferent nerve based on Mott memristors. The stimuli intensity was converted to spike frequency via artificial spiking somatosensory system. To show the feasibility of the system, the author used a piezoelectric device to generate input voltage and showed the frequency response of the system under different pressure intensities. The reviewer believes that researchers in this field will be very interested in the contents of this manuscript. Therefore, the reviewer would recommend the publication of this paper to Nature Communication after the following minor revision. These comments are not to criticize but to strengthen the manuscript.

Response: We thank the referee for this comment. Our responses to the comments one by one are shown as follows.

Comments:

1. Relaxation state can be thought of as a state with an open circuit that has no inputs. However, in the manuscript, current still flows in the relaxation process. The reviewer recommends to explain what each state means in the manuscript.

Response: We thank the referee for this comment. We agree with the referee that the relaxation state could be considered as a state with no inputs. The test for a single device is always operated like that, like reference R3 and R4 do. In these works, the relaxation time is thought of the time that the device back to its high resistance state (HRS) from a low resistance state (LRS) under a small read voltage. In this work, the NbO_x device works in the ASAN circuit with the input voltage. The voltage on the capacitor determines whether the device turns on or turns off. Once the voltage surpasses the threshold voltage of the NbO_x device, the device turns on, and the capacitor discharges through the device to a hold voltage value of the NbO_x device. Here, we considered the relaxation time as the discharge time during which the NbO_x device returns to an HRS from an LRS. As the referee mentioned, the current still flows in this process due to a voltage always exists during working. A relaxation process that completely discharges the capacitor is shown in Fig. R4a. When the ASAN has no input, the capacitor gradually discharges, and the current eventually reaches zero. To clearly clarify this process, we modified the manuscript on page 7 as “Here, for easy understanding, we define the charging time from V_H to V_{TH} as the integration time and the discharging time from V_{TH} to V_H as the relaxation time.” In addition, the integration and relaxation processes are indicated in Fig. 3b, as shown in Fig. R5b.

Fig. R4. (a) The complete relaxation process of the capacitor with no input. (b) (Fig. 3b). Zoom-in views of (a). The charging time from V_H to V_{TH} is defined as the integration time and the discharging time from V_{TH} to V_H as the relaxation time.

2. What is the device's intrinsic parasitic capacitance?

Response: We thank the referee for this comment. We performed the capacitance test on the device after a calibration operation of the Agilent B1500A C-V test module, as shown in Fig. R5. The results show that the intrinsic parasitic capacitance of the device under OFF state is about dozens of fF.

Fig. R5. The C-V test of the device. Dozens of fF capacitance values under the OFF state of the NbO_x device are observed. The capacitance peak value (~100 pF) is because the increased transient switching current makes the measurement value increased. In this switching moment, the tested value is not the real capacitance.

3. The reviewer recommends to add recently published "Science Advances Vol. 4, no. 11, eaat7387" as a reference which could be related to the article.

Response: We thank the referee for this comment. We add this reference on page 3 as reference [38] and the number of the references is updated accordingly in the revised manuscript.

38. Y. Lee, J. Y. Oh, W. Xu, O. Kim, T. R. Kim, J. Kang, Y. Kim, D. Son, J. B.-H. Tok, M. J. Park, Z. Bao, and T.-W. Lee, Stretchable organic optoelectronic sensorimotor synapse, *Sci. Adv.* **4**, 7387 (2018).

4. In reviewer's opinion, it might be better to add external parasitic capacitor's capacitance in Fig 3. or in the caption.

Response: We thank the referee for this comment.

We added the capacitance value in the annotation of Fig. 3. "(a) Schematic of the ASAN. A resistor (75 kΩ) and a NbO_x memristor with certain parallel parasitic capacitance are combined together ($R_{HRS} \gg R_c \gg R_{LRS}$). In real applications, the voltage on NbO_x top electrode serves as the output spiking signal. The parasitic capacitor ($C_{parasitic}$) is tested to be about 20 pF."

5. The reviewer recommends to plot current that flows at memristor and calculate joule heat value in case of the large input voltage. Moreover, please check whether the value of joule heat is enough to make ASAN dwell on its "on-state".

Response: We thank the referee for this comment. In the case of 6.2 V input voltage, according to Kirchoff's voltage law, we could obtain that $V_{in} * R_{on} / (R_{on} + R_c) = V_{out}$ when the NbO_x device turns on, the schematic is shown in Fig. R6a. Thus, when ASAN dwells on "on-state" the resistance value of the NbO_x device is equal to $R_{on} = V_{out} * R_c / (V_{in} - V_{out}) = 1.72 \text{ V} * 75 \text{ k}\Omega / (6.2 \text{ V} - 1.72 \text{ V}) = 28.8 \text{ k}\Omega$. Here, the high resistance value of the NbO_x is roughly evaluated to be 35 M Ω according to Supplementary fig. S1. According to these two resistance values, we calculated the current flowing through the NbO_x device and the power consumption on the device under a 5.6 V input voltage, as shown in Fig. R6b. The results show that when the power on NbO_x device is lower than 100.6 μ W, the device returns back to its high resistance state (HRS). Due to the nonlinear I-V characteristics of the NbO_x device, the resistance value at the moment of returning to HRS should be higher than 28.8 k Ω . Thus, the real lowest power consumption that could hold the device's "on-state" should be less than 100.6 μ W.

The current flowing through the NbO_x device and the power consumption on the device under a 6.2 V input voltage is shown in Fig. R6c. About 103 μ W power consumption is observed when the device holds on its "on-state", which is larger than 100.6 μ W. We believe this power consumption is sufficient to hold the device on "on-state", thus makes ASAN dwell on its "on-state". Furthermore, compared output voltages under 5.6 V/6.0 V input voltages, the output voltage of the ASAN under 6.2 V is always larger than the hold voltage of the device after the NbO_x device turning on, as shown in Fig. R6d. Thus, the

oscillation behavior cannot be obtained anymore. To clearly declare this phenomenon, we modified the main text by adding the following sentences on page 10: “It should be noted that the ASAN stops firing continuously under a higher voltage owing to the voltage divided on NbO_x device is always larger than its holding voltage after the first-fire event. The voltage on the device could generate sufficient Joule-heat to hold the device in its “on-state” (see fig. S9).”

Fig. R6. (a) The schematic of the ASAN. In the case of 6.2 V input voltage, according to Kirchoff's voltage law, $R_{on} = V_{out} * R_c / (V_{in} - V_{out}) = 1.72 \text{ V} * 75 \text{ k}\Omega / (6.2 \text{ V} - 1.72 \text{ V}) = 28.8 \text{ k}\Omega$. The HRS value is estimated to be about $1 \text{ M}\Omega$. (b) The calculated transient current flowing through the NbO_x device and the power consumption on the device under a 5.6 V input voltage of the ASAN. $100.6 \mu\text{W}$ hold power value is observed, which

should be smaller in real due to the nonlinear I-V characteristics of the NbO_x device. (c) The calculated transient current flowing through the NbO_x device and the power consumption on the device under a 6.2 V input voltage of the ASAN. About 106 μW hold power is observed, which is enough to hold the device in its “on-state”. (d) Comparison of the oscillation behavior under 5.6 V, 6.0 V, and 6.2 V input voltages, respectively. The output voltage of the ASAN under 6.2 V input voltage is always larger than the V_H of the NbO_x device, which results in the device dwells in its “on-state”.

6. It seems that, when the input voltage is large, the spiking frequency descends initially and stop spiking. But when does the ASAN stops spiking? In Fig 4c. it says that when the voltage is 6 V, the frequency remains in about 6 kHz. However, in Fig 4f., the ASAN stops spiking when the voltage is 6 V, which should have the frequency of 6kHz according to Fig 4c. Please describe the occasion that ASAN stops spiking in Fig 4. or in the manuscript.

Response: We thank the referee for this comment. During our test, the ASAN stops spiking in Fig. 4c when the input voltage is 6.2 V. Under 6 V input voltage, the ASAN still works but nearly stops. The difference of stop spiking voltages between Fig. 4c and Fig. 4f is resulted from the variability of the hold voltage, as shown in Fig. R7a. Fig. R7b shows the evolution of the V_{TH} and V_H during DC operations. A discrete distribution in both V_{TH} and V_H is observed. It should be noted that the device in Fig. R7a and R7b is not the same one as that in Fig. 4, so there are differences between both V_{TH} and V_H.

As we analyzed in comment 5, the stop firing voltage depends on the V_H. After the first firing, if the voltage divided on NbO_x device is always larger than V_H, the device stays in

its “on-state” and no fire again. The V_H during the test time of Fig. 4c and Fig. 4f are 1.71 V and 1.62 V, respectively, as shown in Fig. R7c and R7d. When we tested Fig. 4f, a smaller voltage than 6 V resulted in the voltage divided on NbO_x device always larger than 1.62 V. Thus, the stop spiking voltage is slightly lower than that in Fig. 4c. Here, the stop spiking voltage is about 5.9 V.

To describe the occasion of stopping firing, except for the added sentence for comment 5 to explain the stop voltage, we also modified the main text on page 11 as follows: “We note that the stop spiking voltage in Fig. 4f is about 5.9 V, which is slightly smaller than that in Fig. 4c (6.2 V). This difference results from the V_H variability of NbO_x device. The V_H in the time of testing Fig. 4f (1.62 V) is lower than that in Fig. 4c (1.71 V), thus a lower stop spiking voltage in Fig. 4f is observed.”

Fig. R7. (a) 50 cycles I-V curve of the NbO_x device (b) Evolutions of the V_{TH} and V_H during DC cycles operation. A segmental distribution in both V_{TH} and V_H is observed. (c) and (d) The V_H during the test time of Fig. 4c and Fig. 4f are 1.71 V and 1.62 V, respectively. It should be noted that the device in Fig. R7a and R7b is not the same one as that in Fig. 4, so there are differences between both V_{TH} and V_H .

7. Please check over some errors due to a typo. In page 11 line 265, the word "verve" should be changed to "nerve".

Response: We thank the referee for this comment. We have corrected it in the main text.

“These results suggest that our afferent **nerve** has a great potential to be used in spiking neurorobotics.”

References

- R1 M. D. Pickett and R. S. Williams, Sub-100 fJ and sub-nanosecond thermally driven threshold switching in niobium oxide crosspoint nanodevices, *Nanotechnology* 23, 215202 (2012).
- R2 S. Kumar, J. P. Strachan, and R. S. Williams, Chaotic dynamics in nanoscale NbO₂ Mott memristors for analogue computing, *Nature* 548, 318-321 (2017).
- R3 Z. Wang, S. Joshi, S. E. Savel'ev, H. Jiang, R. Midya, P. Lin, M. Hu, N. Ge, J. P. Strachan, Z. Li, Q. Wu, M. Barnell, G. L. Li, H. L. Xin, R. S. Williams, Q. Xia, and J. J. Yang, "Memristors with diffusive dynamics as synaptic emulators for neuromorphic computing", *Nat. Mater.* 16, 101-108 (2016).
- R4 X. Zhang, S. Liu, X. Zhao, F. Wu, Q. Wu, W. Wang, R. Cao, Y. Fang, H. Lv, S. Long, Q. Liu, and M. Liu, "Emulating short-term and long-term plasticity of bio-synapse based on Cu/a-Si/Pt memristor", *IEEE Electron Device Lett.* 38, 1208-1211 (2017).

REVIEWERS' COMMENTS:

Reviewer #1 (Remarks to the Author):

Acceptance is suggested.

Reviewer #2 (Remarks to the Author):

In the manuscript titled "An artificial spiking afferent nerve based on Mott memristor for neuro-robotics", the authors investigated artificial spiking afferent nerve based on Mott memristors. The stimuli intensity was converted to spike frequency via artificial spiking somatosensory system. The reviewer believes that researchers in this field will be very interested in the contents of this manuscript. The manuscript is strengthened after the revision and can be accepted by Nature Communication now.

We thank all the reviewers for their precious time and positive comments on our manuscript.

Reviewer #1 (Remarks to the author)

Acceptance is suggested.

Response: We thank the referee for the positive comments on the revised manuscript.

Reviewer #2 (Remarks to the author)

In the manuscript titled “An artificial spiking afferent nerve based on Mott memristor for neuro-robotics”, the authors investigated artificial spiking afferent nerve based on Mott memristors. The stimuli intensity was converted to spike frequency via artificial spiking somatosensory system. The reviewer believes that researchers in this field will be very interested in the contents of this manuscript. The manuscript is strengthened after the revision and can be accepted by Nature Communication now.

Response: We thank the referee for the positive comments on the revised manuscript.